# Hierarchical composition of reliable recombinase logic devices

Sarah Guiziou[1], Pauline Mayonove[1] & Jerome Bonnet [1]

A major goal of synthetic biology is to reprogram living organisms to solve pressing challenges in manufacturing, environmental remediation, and healthcare. Recombinase devices can efficiently encode complex logic in many species, yet current designs are performed on a case-by-case basis, limiting their scalability and requiring time-consuming optimization. Here we provide a systematic framework for engineering reliable recombinase logic devices by hierarchical composition of well-characterized, optimized recombinase switches. We apply this framework to build a recombinase logic device family supporting up to 4-input Boolean logic within a multicellular system. This work enables straightforward implementation of multicellular recombinase logic and will support the predictable engineering of several classes of recombinase devices to reliably control cellular behavior.

[1] Centre de Biochimie Structurale, INSERM U1054, CNRS UMR5048, Université de Montpellier, 29 rue de Navacelles, 34090 Montpellier, France. Correspondence and requests for materials should be addressed to J.B. (email: jerome.bonnet@inserm.fr)

The field of synthetic biology aims at programming cellular and organismal behavior to address pressing challenges and answer basic research questions[1]. To do so, synthetic biologists took inspiration from electronic designs to engineer logic gates operating in living cells[2–5]. Gates were built using transcriptional regulators[6,7], RNA molecules[2,8,9], or site-specific recombinases[10,11]. Recombinase logic is of particular interest because of its compact design, modularity, portability (works in many species), and associated memory. Recombinase logic gates operate by specifically inverting or excising DNA sequences containing regulatory elements flanked by recombination sites. Because recombination reactions are irreversible, recombinase logic devices are single use (one-shot) and belong to the class of asynchronous logic devices: they can respond to multiple signals even if these are not occurring at the same time. While not implementing *stricto sensu* combinatorial logic, the flexibility and single-layer architecture of recombinase logic devices are advantageous in a number of situations for which reversibility and synchronous response are not required. For example, the memory associated with the system can be extremely useful for applications requiring end-point measurements, like biosensing and diagnostics[12].

Recombinase logic gates present different arrangements of recombination sites and regulatory elements. Such architectures can produce devices having highly variable behavior[13,14]. Device complexity increases with the numbers of inputs to be computed, and engineering recombinase logic gates is still a trial-and-error process. Scaling-up recombinase logic circuits and extending their applications therefore requires simple and accessible engineering frameworks.

One approach to simplify logic design is to use Distributed Multicellular Computation (DMC)[4,7,15]. As in natural ecosystems, the computational labor is divided between various strains that perform specific tasks. In addition to reducing metabolic burden and supporting the reuse of biological parts, DMC is highly modular by nature and permits the implementation of all combinatorial logic functions by mixing different strains chosen from a reduced strain library[7,16].

We recently devised a strategy for distributed multicellular recombinase logic[17]. In our approach, the logic equation is written as a sum of product of literals corresponding to the Canonical Disjunctive Normal Form (CDNF) obtained using the Quine McCluskey algorithm[18]. Each term of the CDNF implements a portion of the whole function, termed a subfunction, and is performed by a particular strain within a multicellular system (Fig. 1a). Each subfunction (i.e., a product of NOT and IDENTITY functions) is executed by a particular recombinase logic device. For a given number of inputs, a fixed, small number of recombinase devices can be differentially combined to obtain all combinatorial logic functions. The advantages conferred by this modular design are well exemplified by the number of functions that can be implemented using a reduced number of devices. For instance, all 65536 4-input logic functions can be implemented using only 14 standardized recombinase logic devices.

Recombinase logic devices are obtained by hierarchically combining two classes of logic elements, ID and NOT, according to a specific set of rules[17] (Fig. 1b). ID elements implement the IDENTITY function, in which the output is ON when the input is present. In recombinase operated ID-elements, excision of a terminator triggers gene expression. NOT elements implement the NEGATION function, so that the output is ON when the input is absent and vice-versa. In NOT elements, recombinase-mediated excision of a promoter turns-off gene expression.

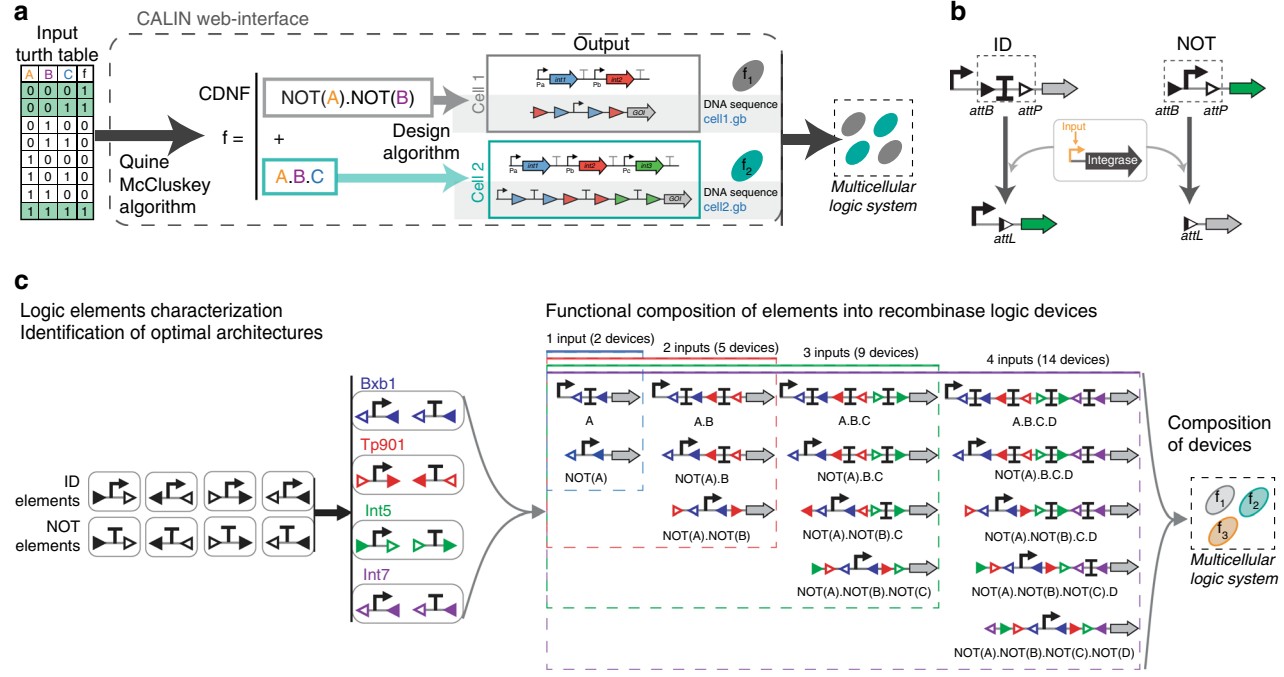

**Fig. 1** A hierarchical composition framework for recombinase logic design and engineering. **a** Design pipeline for multicellular recombinase logic. **b** Logic devices are built from NOT and IDENTITY (ID) elements, respectively composed by being nested or placed in series[17]. NOT elements are composed of a promoter surrounded by integrase sites. In presence of the integrase, the promoter is excised and expression of the output gene is switched from ON to OFF. IDENTITY elements are composed of a terminator surrounded by integrase sites. In presence of the integrase, the terminator is excised, and gene expression is switched ON. **c** Workflow for engineering reliable recombinase logic devices. ID and NOT elements with all possible integrase-site orientations and permutations are characterized (black triangle: attB site, white triangle: attP site). Elements with the best behavior are selected and composed to design the 14 logic devices required to compute all 4-input Boolean functions. For recombinase logic devices, blue sites (Bxb1 sites) correspond to input A, red sites (TP901-1) to input B, green sites (Int5) to input C and purple sites (Int7) to input D

In our design, assembling higher-order recombinase devices relies on successful functional composition of several ID and NOT elements each responding to a different enzyme and signal (Fig. 1c). Here we show that orientations and positions of recombination sites can greatly affect logic element behavior and downstream gene expression. We identify well-operating elements from a combinatorial library and compose them into recombinase logic devices operating reliably. Finally, we demonstrate that these devices can be used to implement complex logic functions within a bacterial community. We anticipate multicellular recombinase logic to support many applications in bioprocessing, healthcare and material sciences.

## Results

**Identification of well-operating logic elements**. We aimed at identifying well-operating elements from a library containing ID and NOT elements with all possible permutations and orientations of integrase attachment (*att*) sites.

We characterized elements responding to four orthogonal serine integrases, Bxb1, TP901-1, Int5, and Int7[13,19] (Supplementary Fig. 1). All logic elements were based on the same scaffold composed of a strong promoter (P7)[20], a ribozyme (RiboJ) to obtain an identical 5′UTR in all constructs[21], a bicistronic RBS (BCD) to prevent interactions between the RBS and the coding sequence[20], and a superfolder green fluorescent protein (sfGFP) as a reporter[22]. To speed up the process, we directly synthesized all constructs corresponding to the element before and after recombination occurred.

We measured GFP fluorescence intensity in the different states for all constructs, and observed important variations of element behavior depending on integrase-site positions and orientations (Fig. 2 and Supplementary Figs. 2, 3, and 4). Two NOT elements responding to TP901-1 integrase had leaky GFP expression (six times above the negative control) in their supposedly OFF state (Fig. 2a), suggesting directional cryptic promoter activities in TP901-1 *att* sites. ID-elements also had important differences

depending on integrase-site orientations, especially for non-recombined elements, with up to 100-fold difference in gene expression between Int5 elements (Fig. 2b). For TP901-1, we observed that all ID-elements expressed GFP in their OFF states and tracked the problem to inefficient termination activity. We obtained a well-operating ID-element for TP901-1 by using a different terminator (Fig. 2b). For each integrase and function, we selected logic elements having low leakage in their OFF states and high switching fold changes, ranging between 222 and 649, except for TP901-1 ID-element which has an 83-fold change.

**Functional composition of logic elements**. We then composed these elements to obtain fourteen devices capable of implementing all 4-input logic functions[17]. We characterized the response of these devices to all possible input combinations. To streamline our characterization process, we decoupled recombination from any particular control signal by co-transforming the logic devices with various constitutive gene expression cassettes containing all possible combinations of the four integrases (Fig. 3, Supplementary Fig. 5).

All logic devices behaved as expected with very distinct ON and OFF states, demonstrating the possibility to obtain devices operating reliably from well-characterized logic elements (Fig. 4). On a quantitative point of view, most devices exhibited slightly lower expression levels in their ON states compared to the positive control (i.e., promoter-rbs-GFP). This attenuation is certainly caused by the residual *att* sites positioned between the promoter and the reporter. More importantly, some devices had an expression level of GFP above background in their OFF states ("leakage"). This leakage is likely due to a reduced termination strength for some intermediate recombination states in which one or multiple terminators from a serie have been excised.

In order to quantify device performances, we defined a "maximum leakage" value as the highest median fluorescence intensity value measured across all OFF states (Supplementary Fig. 6). We used this conservative approach to calculate device

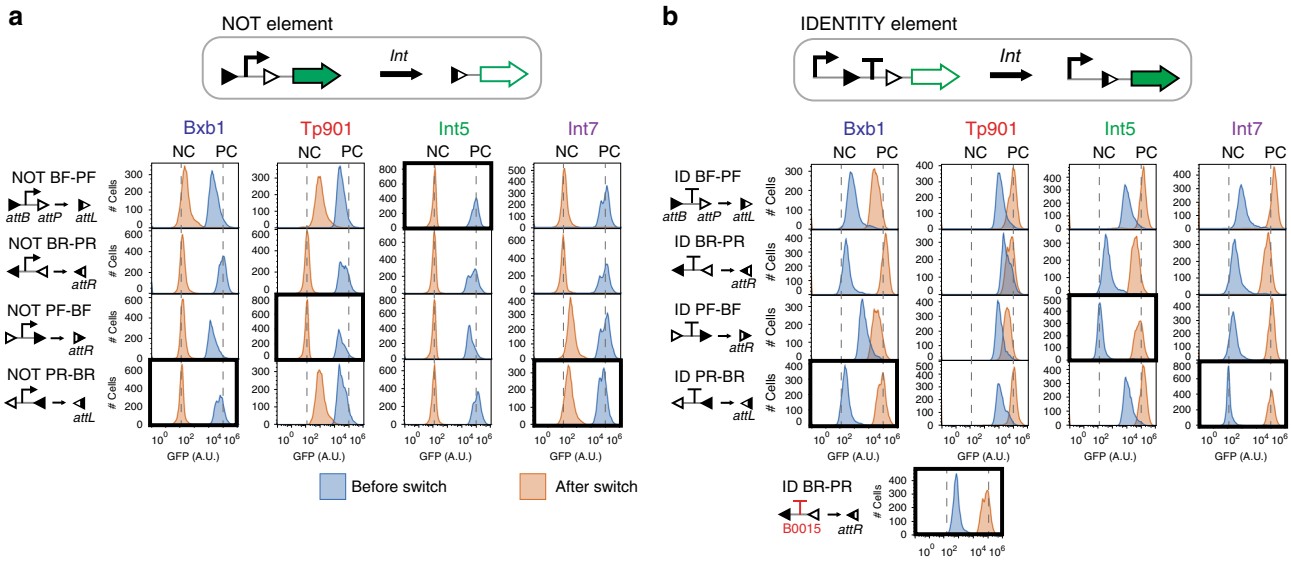

**Fig. 2** Identification of well-operating logic elements form a library of NOT and IDENTITY elements. NOT and IDENTITY elements responding to four integrases (Bxb1, TP901-1, Int5, Int7) were characterized. For each element, four different designs are possible (BF-PF, BR-PR, PF-BF, PR-BR). We measured gene expression before (NO INT, blue) and after switching (INT, orange) by flow cytometry using sfGFP as output. **a** NOT elements characterization and **b** ID elements characterization. Cells were grown in LB with appropriate antibiotics for 16 hours at 37 °C. Boxes indicate the construct that was ultimately chosen. A functional TP901-1 ID-element was obtained by replacing the original terminator (lower insert). 3 experiments were performed in 3 different days with 3 replicates per experiment. A representative example is depicted here. Fold change measurements can be found in supplementary Figs. 2, 3 and 4. Source data are provided as a Source Data file. Int: integrase

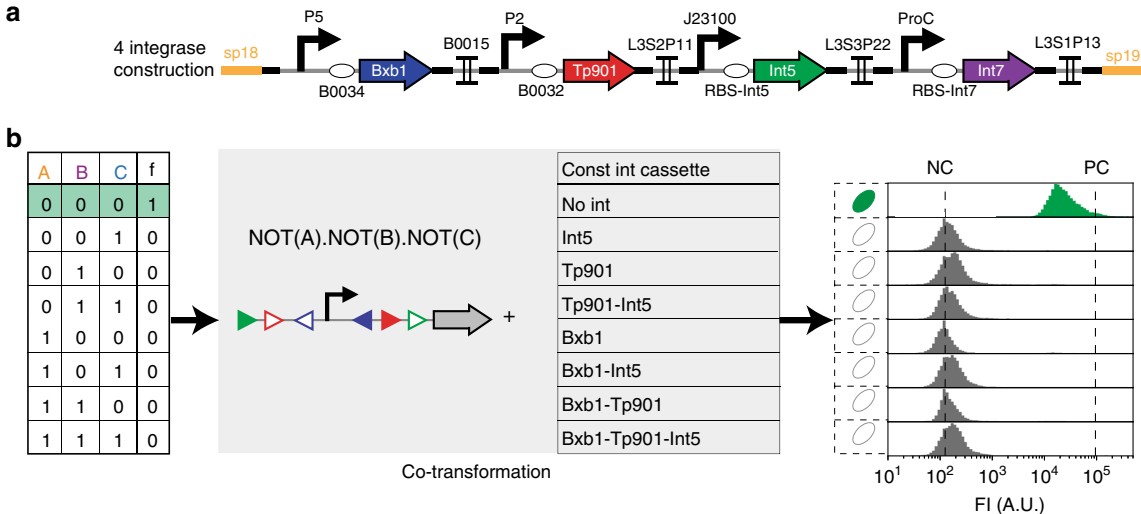

**Fig. 3** Characterization of recombinase logic devices using constitutive integrase expression cassettes. **a** Architecture of the constitutive integrase expression plasmid. Each integrase is driven by a different promoter and RBS. Individual integrase expression cassettes are separated by transcriptional terminators. The 4 integrase generator plasmid is represented here. We generated all possible integrase combination for a total of 16 plasmids (see Supplementary Fig. 5 for characterization data). P promoter, RBS Ribosome Binding Site, Int integrase. **b** We characterized recombinase logic devices response for each input state by co-transforming each device with our combinatorial collection of constitutive integrase cassettes (see materials and methods for details). FI: fluorescence intensity

**Fig. 4** Functional composition of logic elements into recombinase logic devices. For each device, we measured the fluorescence intensity for each state by flow cytometry. Corresponding input states are on the x-axis (0: no input, 1: input). Green histograms correspond to the input states expected to be ON. Fluorescence intensity levels for the negative control (NC, no promoter) and positive control (PC, promoter-RBS-GFP) are represented by lower and upper dash lines, respectively. 2 experiments were performed in 2 different days with 3 replicates per experiment. A representative example is depicted here. Fold change measurements can be found in supplementary Fig. 5. Source data are provided as a Source Data file

fold changes which span from ~30 to 300 fold between OFF and ON state (Table 1, Supplementary Fig. 7). Additionally, all logic devices had a common output threshold between OFF and ON states within 8-fold change, an essential parameter for a multicellular system containing multiple recombinase devices operating together. Because of their standardized architecture, these recombinase logic devices can be easily tuned by changing the transcription input signal (Supplementary Fig. 8, Supplementary Table 1).

**Prototyping multicellular recombinase logic**. We then aimed at prototyping multicellular logic systems relying on multiple recombinase devices operating in concert. The fact that the system is considered in ON state if at least one strain is ON leads to two challenges. First, the signal strength of all recombinase devices in the ON state must be sufficiently high to be detectable even if a subfraction of the population is ON. Second, the potential gene expression leakage observed in certain OFF states (Fig. 4, Table 1) must be low enough so that the multicellular system does not produce false positives. Different strains

### Table 1 Recombinase device characteristics

| # Logic device | # Gate fold change | # Maximum leakage |
|---|---|---|
| A.B | 58 | 1.9 |
| not(A).B | 300 | 1.8 |
| not(A).not(B) | 169 | 1 |
| A.B.C | 86 | 3.2 |
| not(A).B.C | 42 | 11 |
| not(A).not(B).C | 34 | 3.2 |
| not(A).not(B).not(C) | 112 | 1.5 |
| A.B.C.D | 37 | 4.8 |
| not(A).B.C.D | 55 | 11 |
| not(A).not(B).C.D | 36 | 1.9 |
| not(A).not(B).not(C).D | 60 | 1.2 |
| not(A).not(B).not(C).not(D) | 34 | 4.5 |

The gate fold change corresponds to the fold change between the ON state and the maximum OFF state, equivalent therefore to a minimum fold change. The maximum leakage corresponds to the fold change of the maximum OFF state

expressing various levels of GFP might also exhibit differences in growth rates resulting in the disappearance of some sub-population over time.

To prototype our system, we co-transformed logic devices with the different constitutive integrase cassettes and then mixed different strains in various states to obtain a multicellular system simulating all possible input combinations (Fig. 5). We built a two-strain system for 3-input logic and a three strain system for 4-input logic (see methods for details). We measured the fluorescence intensity of the multicellular system in each input state and were able to clearly distinguish expected ON and OFF states for all systems. We did observe differences in ON state intensities, some directly related to the differences of ON level of the separated devices, others to differences in growth rates between strains having different GFP or integrase expression status (Supplementary Fig. 9). We also obtained multicellular systems with more constant output levels by using devices having lower transcription input signals, albeit at the cost of a lower fold change (Supplementary Fig. 10). Taken together, these data unambiguously demonstrate the feasibility of composing strains containing various recombinase devices to implement complex Boolean functions at the multicellular level.

## Discussion

In this work we have demonstrated that standardized, optimized recombinase logic elements can be hierarchically composed into higher-order recombinase logic devices that reliably behave as predicted. Predictable, hierarchical composition of simple logic elements is compatible with other genetic designs using recombinases[13,17,23–26] and will thus extend the robustness and range of applications of this highly useful class of synthetic circuits.

Here we provide fourteen recombinase logic devices having a common output threshold and that can be combined to implement complex logic functions within multicellular systems. Multicellular logic designs can be obtained using our automated design webserver CALIN[17] (http://synbio.cbs.cnrs.fr/calin/). Because recombinase activation is decoupled from logic gate operation, these logic devices could be directly reapplied for many purposes. Examples of applications range from industrial processes using microbial consortia[27] (e.g., bioprocessing, drug production, bioremediation) to the engineering of synthetic

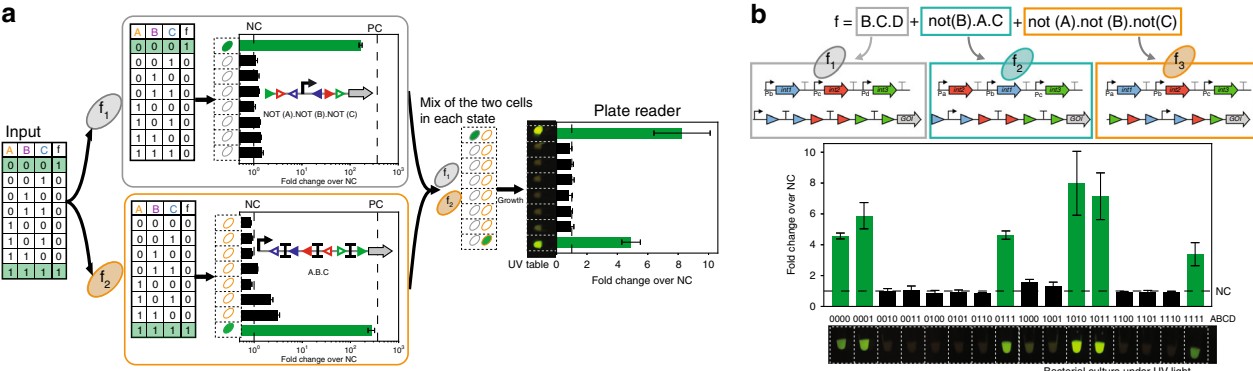

**Fig. 5** Prototyping multicellular recombinase logic. **a** Workflow for multicellular logic system prototyping. The consensus 3-input function is decomposed in two sub-functions implemented using two logic devices. To prototype this logic system for each input state, we mixed the two strains containing the recombinase logic device and different constitutive integrase cassettes corresponding to the different input states. After overnight growth, we measured the bulk fluorescence intensity of the whole-population using a plate reader. Bar graphs corresponds to the fold change in GFP median fluorescence intensity over the negative control. NC Negative Control (no promoter), PC positive control (promoter-RBS-GFP). Data are from two experiments performed in different days with three replicates per experiment. Error bars: ±s.d. The photograph correspond to three co-culture replicates for each state centrifuged together, resuspended in 20 µL and observed under a UV light. **b** Prototyping a 4-input, 3-strain multicellular logic system. The bar graphs and pictures were obtained as in (**b**). Source data are provided as a Source Data file

microbial communities for healthcare[28]. Multicellular recombinase logic with built-in memory would also be useful to engineer complex cellular systems capable of self-organization and differentiation, with applications to material science[29] and tissue engineering[30,31].

## Methods

**E. coli strains and media**. DH5alphaZ1[32] E. coli strain was used in this study (laciq, PN25-tetR, SpR, deoR, supE44, Delta(lacZYA-argFV169), Phi80 lacZDeltaM15, hsdR17(rK- mK + ), recA1, endA1, gyrA96, thi-1, relA1). E. coli was grown on LB media with antibiotic corresponding to the transformed plasmid(s). Antibiotics were purchased from Sigma and used at the following concentration: chloramphenicol: 20 μg mL$^{-1}$, kanamycin: 25 μg mL$^{-1}$, carbenicillin: 50 μg mL$^{-1}$ (for ampicillin resistance). For co-transformation of two plasmids, the two corresponding antibiotics were used at the previously defined concentration divided by two.

**Molecular biology**. We used vectors pSB4K5 and J66100 (from parts.igem.org). The pSB4K5 plasmid containing a kanamycin resistance cassette and a pSC101 low-copy origin of replication was used for cloning of BP and LR targets, parts, logic elements, and recombinase logic devices. The J66100 plasmid is a derivative of J64100 in which the chloramphenicol resistance cassette was replaced by an ampicillin resistance one. J66100 has a regulated ColE1 origin of replication and was used the cloning integrase cassettes.

All plasmids used in this study were derived from these two vectors and fragments were assembled using one-step isothermal assembly following standard molecular biology procedures[33]. Enzymes for the one-step isothermal assembly were purchased from New England BioLabs (NEB, Ipswich, MA, USA). PCR were performed using Q5 PCR master mix and One-Taq quick load master mix for colony PCR (NEB), primers were purchased from IDT (Louvain, Belgium), and DNA fragment from Twist Bioscience. Plasmid extraction and DNA purification were performed using kits from Biosentec (Toulouse, France). Sequencing was realized by GATC Biotech (Cologne, Germany).

**Construction of BP and LR targets**. For Tp901 and Bxb1 targets, the BP and LR targets from Bonnet et al.[23] were used. For Int3, Int4, Int5, and Int7 targets, a template sequence composed of mKate in forward orientation and GFP in reverse orientation was synthesized and assembled in pSB4K5 between sp0 and spN into the pSB4K5 vector. Then, target fragments containing the sequence between the mKate and GFP coding sequences were synthesized and assembled in the previously constructed template sequence.

**Construction of parts, elements and devices**. We use as a backbone for logic elements and devices the expression operating unit from Guiziou et al.[34], which contains several spacers optimized for Gibson assembly. The construct was inserted in pSB4K5 (see DNA sequence supplementary file for insertion locus). For the construction of NOT-, IDENTITY elements, and positive and negative controls, the previous construct was used as a template and amplified between sp0 spacer and the beginning of the GFP for one-step isothermal assembly with linear fragments corresponding to each element. For logic devices, the terminator in 3′ of the construct was switched from B0015 to L3S3P00.

**Construction of integrase cassettes**. A cassette with each Integrase under the control of lac promoter was synthetized and cloned in J66100 plasmid. These cassettes were used to characterize integrase function and orthogonality (Supplementary Fig. 1). To build a combinatorial library of constitutively expressed integrases (Supplementary Fig. 4), we first synthesized a landing pad composed of promoters, terminators, and spacers and cloned it in J66100 (see DNA sequence file for insertion locus). Each integrase was then amplified from the previous Plac construct and inserted separately in the landing pad generated single integrase cassette. All integrase cassettes variants were then built through gibson assembly by combination of these single integrase cassettes.

**Flow-cytometer measurements**. Quantification of expression levels in all strains was performed using an Attune NxT flow-cytometer (Thermofisher) equipped with an autosampler. Experiments were performed on 96 wells plates with three replicates per plates. For flow-cytometry measurements, 20,000 bacteria events were analysed. A gate was previously designed based on forward and side scatter graphs to remove debris from the analysis. GFP fluorescence intensity was measured using excitation by a 488 nm laser and a 510/10 nm filter (BL1). RFP excitation was performed by a 561 nm laser and filter 615/25 nm (YL2). Voltages used were FFS: 440, SSC: 340, BL1: 360, for all experiments except with BP and LR targets, and BL1: 400 and YL2: 400, for experiments with BP and LR targets. Data were analyzed and presented using the Flow-Jo (Tristar) software.

**Characterization of elements**. A glycerol stock from each construct was streaked on LB agar plate supplemented with kanamycin. 96 deep well plates filled with 500 μL of LB with kanamycin antibiotic were inoculated with three clones from the freshly streaked plates. For all experiments, three clones of the negative control strain corresponding to RBS-GFP without promoter and the positive control strain

corresponding to P7-RBS-GFP were inoculated. Plates were grown 16 hours at 37 °C. Cultures were diluted 40 times on Focusing Fluid and measured on flow-cytometer.

Three experiments with three replicates per experiments were performed for elements, integrase sites, and terminators characterizations. Data were analyzed using Flow-Jo. Bacteria events were gated to remove debris from the analysis by plotting FSC-H over SSC-H. For Fig. 2, the histogram of GFP fluorescence intensity (BL1-H) of one representative replicate is represented. Additionally, for each independent experiment, the median GFP fluorescence intensity of the bacterial population for each replicate was extracted, corresponding to BL1-H median and the fold change over the NC control was calculated. In supplementary figures, the mean of fold change between the three experiments is represented, and the error bar corresponds to the standard deviation between the three experiments.

**Characterization of integrases cassettes**. For integrase characterization, each Plac-integrase plasmids and dual controller for Tp901 integrase[13] was co-transformed with BP targets. For constitutive integrase cassette characterization, each constitutive integrase cassette was co-transformed with the BP targets corresponding to the integrase that it should express.

For both experiments, 96 deep wells plate filled with 500 μL of LB per well were inoculated with three clones per co-transformation and three clones per control corresponding to the BP target and LR target strains. For integrase characterization with Plac-integrase plasmid and dual controller plasmid, LB was supplemented with 100 μM of IPTG for co-transformation with Plac-integrase and 1% of Arabinose for co-transformation with the dual controller for expression of Tp901. Plates were grown 16 hours at 37 °C. Cultures were diluted 40 times on Focusing Fluid and directly measured on flow-cytometer according to previously described methods.

Data analysis was performed using Flow-Jo. Bacteria events were gated to remove debris from the analysis by plotting FSC-H over SSC-H. Data were represented using a density plot of BL1-H over YL2-H, corresponding to the GFP fluorescence intensity over the RFP fluorescence intensity. The proportions of bacteria in BP or LR states were obtained using BL1-H over YL2-H plot by gating the population corresponding to the BP or LR target strain. Data represented in the heatmap correspond to the mean of the proportion obtained with the three replicates in one experiment. Details about the flow-cytometry gating strategy can be found in supplementary Fig. 11.

**Characterization of recombinase logic devices**. Each device was co-transformed with each integrase cassettes corresponding to its input number. For each transformation, three clones were picked in 500 mL of LB in 96 deep well plate. Additionally, the negative control (RBS-GFP without promoter) strain and positive control (Promoter-RBS-GFP) strain were streaked from glycerol stocks and three clones were picked and inoculated. Plates were grown 16 hours at 37 °C. Cultures were diluted 40 times on Focusing Fluid and measured on flow-cytometer. Two experiments with three replicates per experiments were performed. Data were analyzed using Flow-Jo using the same procedure than the one detailed previously for element characterizations.

**Multicellular logic system prototyping**. Devices were co-transformed with corresponding constitutive integrase cassettes. Three clones per transformation were inoculated in 500 μL of LB in 96 deep well plate. Plates were incubated during 16 hours at 37 °C to reach stationary phase. From the stationary phase culture, cells were mixed in identical proportions and diluted 1000 times for growth in 500 μL of LB in 96 deep well plate. Plates were incubated 16 hours at 37 °C. The co-cultures were diluted four times in PBS and analyzed using a plate reader for measurement of bulk fluorescence intensity. Additionally, co-cultures were diluted 200 times in focusing fluid and analyzed on flow-cytometer. Finally, the three replicates were mixed, centrifuged, and the cell pellets were resuspended in 20 μL of PBS in PCR tubes and imaged under UV tables. Plate reader measurement were performed using a BioTek Cytation 3. GFP fluorescence intensity (Excitation: 485 nm, Emission: 528 nm and 85 gain) and absorbance at 600 nm were measured. For each sample, GFP fluorescence intensity over absorbance at 600 nm were calculated and the mean value was calculated between the three replicates. The fold change over the negative control was determined from this mean value over the one of the Negative control. The error bars correspond to standard deviation in fold change. Flow-cytometry experiments were performed as detailed in the corresponding section. To determine the proportion of cells in ON state (expressing GFP), a first gate was performed to select bacteria events using FSC-H over SSC-H density plot. A second gate was performed from bacteria events to select single cells using SSC-A over SSC-H density plot. Finally, from single cell event, BL1-H histogram was plotted and cells with more than 4200 fluorescence intensity in arbitrary units were considered ON to determine the proportion of ON cells using a final gate. This procedure was used to analyse flow-cytometry experiments before and after co-culture growth. Details about the flow-cytometry gating strategy can be found in supplementary Fig. 11.

**Reporting summary**. Further information on experimental design is available in the Nature Research Reporting Summary linked to this article.

## Data availability

Source data for main text figures, along with DNA sequences for all constructs are provided in the Source Data file and Supplementary Data 1, respectively. All other raw data are available from the corresponding author on reasonable request. Plasmids are available from Addgene (Integrase generators kit plasmids: Addgene ID 117029-117044; Recombinase logic devices plasmids: Addgene ID 117007-117028).

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

## Acknowledgements

We thank members of the synthetic biology group and of the CBS, P. Lemaire and P. Hersen for fruitful discussions. Support was provided by an ERC Starting Grant "Compucell", the INSERM Atip-Avenir program and the Bettencourt-Schueller Foundation. S.G. was supported by a Ph.D. fellowship from the French Ministry of Research and the French Foundation for Medical Research (FRM). The CBS acknowledges support from the French Infrastructure for Integrated Structural Biology (FRISBI) ANR-10-INSB-05-01.

## Author contributions

S.G. and J.B designed the project. S.G. and P.M. performed and analyzed the experiments. S.G. and J.B. wrote the manuscript.

## Additional information

**Competing interests:** The authors declare no competing interests.

