## [Peer Review File · Nature Communications]

Reviewers' Comments:

Reviewer #1:

Remarks to the Author:

Here the authors of "Hierarchical composition of reliable recombinase logic devices" create a nice framework for implementing up to 4-input logic functions in *e. coli*. The inputs to the demonstrated functions are four integrases, expressed from a single plasmid. The outputs are the expression or repression of GFP. The best functioning logic elements were chosen based on maximum leakage and fold change from OFF to ON state. Using these elements, the authors were able to combine individual strains containing logic submodules to create more complex functions in a multicellular consortium. Using their framework, the authors were able to create all 14 4-input Combinational logic functions.

In general, the manuscript presents an impressive framework for creating recombinase logic. As it stands now the main body is somewhat lacking in content that in a few cases has been pushed to the supplement. More detail and description of the total work that was done would benefit the manuscript. While the authors do a good job of explaining their characterization and construction process, there is not much focus on what exactly distinguishes this system from other examples of recombinase logic. Increased information on what makes this system unique and perhaps and a demonstration of its uniqueness would benefit the manuscript. Additionally, an aspect of the work that is exciting and potentially novel, the multicellular logic, is not emphasized in the main body. More focus on the multi-strain logic and a potential demonstration of the multi-strain logic's applications would improve the work. Finally, an increased comment in the discussion regarding the possible applications specific to this system is needed.

Other comments:

-Fluorescence data:

The presentation of the data in the manuscript is somewhat sparse. Fluorescence data can be presented in a histogram format, such as in Gander et al. 2017 and Nielsen et al. 2016, to allow the reader to better understand the differences in outputs of the circuit states.

-“For given number of inputs, a fixed, small number of recombinase devices can be differentially combined to obtain all possible logic functions”:

This sentence in line 36 and 37 is not strictly true for the system described in the manuscript. While the ID and NOT logic elements are capable of implementing all 4-input combinational logic functions the system is not capable of implementing sequential logic functions and is therefore not capable of implementing all logic functions. This sentence should be corrected, and a citation should be found so an interested reader can find literature explaining functional completeness.

-Circuit inputs:

The inputs of the circuits are not shown clearly in the main body. Figure S4 a does a nice job of illustrating what the inputs consist of and should be moved into the main body.

-Max Leakage

Some nice work was done to characterize the leakage of individual logic elements and the sentence "Some devices had an expression level of GFP above the background ..." on line 105 should be expanded into a more quantitative discussion on how leakage was calculated and considered when choosing which elements to build circuits from should be added.

-Sentence at line 21 "Synthetic biologists.."

This sentence is worded somewhat awkwardly and could be changed to better communicate how synthetic biologists are inspired by electronics.

Reviewer #2:

Remarks to the Author:

This is a nice paper describing a method to evaluate logic expressions using a recombinase-based approach and a mixture of bacterial strains to implement the OR/disjunction in the DNF-like fashion. The set up of the approach is solid and robust, with up to four inputs being implemented. I am quite happy with the data. The methods seem simpler compared to other efforts in the field. In my view the manuscript is sufficiently interesting to qualify for publication in Nature Communications.

My main comments are in the area of terminology and explanation. What the authors are doing is essentially following the standard DNF construction using a truth table, with each row of the truth table implemented by one strain. The mixture of strains is equated to the OR operation. I could imagine this being useful in one or another application. I would ask the authors to present the concept upfront, present DNF formulae and their "mechanism", and they show their experimental results - it will make much more sense rather than coming out of the blue.

Second, and this is the general problem in the field but nonetheless, these systems are not bona fide combinatorial logic gates because of their memory features to recombinase actions, which are irreversible. There are really state machines with states encoded in the partial recombination products. State machines can simulate logic, so there is no intrinsic problem, but it will be much nicer if the true nature of these systems is discussed upfront and not swept under the carpet.

Lastly, the authors should cite the work by Rinaudo et al, Nature Biotech 2017, as they were the first to introduce normal logic forms to synthetic biology.

To summarize, the paper can be published after the comments above have been addressed.

First, we would like to thank the reviewers for their constructive comments which we believe have improved the quality of the paper.

While this paper was in review, we realized we made a mistake in plotting the data associated with the characterization of elements. We have corrected this error in the revised version. These changes do not affect the conclusions of our work, in particular, they do not affect at all figure 4, showing the characterization of 14 reliable logic devices based on the composition of these well-behaving logic elements. We apologize for the inconvenience caused. We provide a full description of these changes after the point-by-point response.

Below is a full point-by-point response to the reviewer's comments.

Reviewer #1 (Remarks to the Author):

Here the authors of “Hierarchical composition of reliable recombinase logic devices” create a nice framework for implementing up to 4-input logic functions in *e. coli*. The inputs to the demonstrated functions are four integrases, expressed from a single plasmid. The outputs are the expression or repression of GFP. The best functioning logic elements were chosen based on maximum leakage and fold change from OFF to ON state. Using these elements, the authors were able to combine individual strains containing logic submodules to create more complex functions in a multicellular consortium. Using their framework, the authors were able to create all 14 4-input Combinational logic functions.

- As it stands now the main body is somewhat lacking in content that in a few cases has been pushed to the supplement. More detail and description of the total work that was done would benefit the manuscript.

We added some explanation about the theoretical framework, both in the text and in Fig. 1a. We also moved the table presenting fold changes and the explanation on logic devices characterization from the supplement to the main text. These additional details and descriptions indeed enhance the manuscript understandability, thanks for the comment.

- In general, the manuscript presents an impressive framework for creating recombinase logic. While the authors do a good job of explaining their characterization and construction process, there is not much focus on what exactly distinguishes this system from other examples of recombinase logic. Increased information on what makes this system unique and perhaps and a demonstration of its uniqueness would benefit the manuscript. Additionally, an aspect of the work that is exciting and potentially novel, the multicellular

logic, is not emphasized in the main body. More focus on the multi-strain logic and a potential demonstration of the multi-strain logic's applications would improve the work. Finally, an increased comment in the discussion regarding the possible applications specific to this system is needed.

Thanks for the comment. We have added more information in the introduction to discuss the specificities of multicellular logic systems. We also added a paragraph in the discussion to provide a context for applications for multicellular logic systems.

Other comments:

-Fluorescence data:

The presentation of the data in the manuscript is somewhat sparse. Fluorescence data can be presented in a histogram format, such as in Gander et al. 2017 and Nielsen et al. 2016, to allow the reader to better understand the differences in outputs of the circuit states.

We thank the reviewer for this insightful comment! We have followed the advice, and provide a new figure for elements (fig2) and devices (now fig4) characterization that uses flow-cytometry histograms. Indeed, we believe this change really improves data presentation and allows the reader to better grasp the behavior of each device while displaying raw data. As fold change measurements are still an important point, we added in the main text a table summarizing the fold change and maximum leakages for each gate. We moved previous fig 3a in the Supplement.

-“For a given number of inputs, a fixed, small number of recombinase devices can be differentially combined to obtain all possible logic functions”:

This sentence in line 36 and 37 is not strictly true for the system described in the manuscript. While the ID and NOT logic elements are capable of implementing all 4-input combinational logic functions the system is not capable of implementing sequential logic functions and is therefore not capable of implementing all logic functions. This sentence should be corrected, and a citation should be found so an interested reader can find literature explaining functional completeness.

This is right. We changed the sentence to “a small number of recombinase devices can be differentially combined to obtain all possible ***combinatorial*** logic functions”. Regarding functional completeness, while we agree that it is an important concept, our set of devices is not functionally complete *per se*. In fact, we use input permutations to obtain all logic functions. The advantage is that fewer devices need to be characterized and optimized; the drawback is that we need all integrases to be connected to all inputs. We believe providing this explanation in the text would be confusing for the reader and not totally into the scope of this particular article. All details can be found in our previous paper (see Guiziou *et al.*, *ACS Synbio*, 2018).

-Circuit inputs:

The inputs of the circuits are not shown clearly in the main body. Figure S4 a does a nice job of illustrating what the inputs consist of and should be moved into the main body.

Point taken. We have added a new figure (fig.3) in which we describe the integrase expression plasmid as well as the characterization process (both moved in the main text from the supplement).

-Max Leakage

Some nice work was done to characterize the leakage of individual logic elements and the sentence "Some devices had an expression level of GFP above the background ..." on line 105 should be expanded into a more quantitative discussion on how leakage was calculated and considered when choosing which elements to build circuits from should be added.

We added an explanation in the text and also moved the table with the maximum leakage and minimum fold change values in the main text.

-Sentence at line 21 "Synthetic biologists.."

This sentence is worded somewhat awkwardly and could be changed to better communicate how synthetic biologists are inspired by electronics.

We have modified this sentence, we hope it is clearer now.

Reviewer #2 (Remarks to the Author):

This is a nice paper describing a method to evaluate logic expressions using a recombinase-based approach and a mixture of bacterial strains to implement the OR/disjunction in the DNF-like fashion. The set up of the approach is solid and robust, with up to four inputs being implemented. I am quite happy with the data. The methods seem simpler compared to other efforts in the field. In my view the manuscript is sufficiently interesting to qualify for publication in Nature Communications.

My main comments are in the area of terminology and explanation. What the authors are doing is essentially following the standard DNF construction using a truth table, with each row of the truth table implemented by one strain. The mixture of strains is equated to the OR operation. I could imagine this being useful in one or another application. I would ask the authors to present the concept upfront, present DNF formula and their "mechanism", and they show their experimental results - it will make much more sense rather than coming out of the blue.

Actually, each row of the truth table is not implemented by one strain. We do not use the standard DNF but instead, we use the Quine McQuickey algorithm to obtain the **canonical disjunctive normal form (CDNF)**, which is a minimization of the full disjunctive normal form. We agree with the reviewer that it is better to actually explain this concept. We added a figure explaining the theoretical framework to pass from the truth table to the biological design (Fig1 A). We also amended the text to clarify this point.

Second, and this is the general problem in the field but nonetheless, these systems are not bona fide combinatorial logic gates because of their memory features to recombinase actions, which are irreversible. There are really state machines with states encoded in the partial recombination products. State machines can simulate logic, so there is no intrinsic

problem, but it will be much nicer if the true nature of these systems is discussed upfront and not swept under the carpet.

True, that is an important point. We have added a paragraph in the introduction making sure that these particular features of the system are well-described upfront.

Lastly, the authors should cite the work by Rinaudo et al, Nature Biotech 2017, as they were the first to introduce normal logic forms to synthetic biology.

We added the reference-thanks for pointing that out.

Please see correction on the next page.

Dear editors, dear reviewers,

We would like to provide a correction concerning our manuscript. We realized while the manuscript was in review that we made an error while plotting some of the data for figure 2. We believe the main results and message of our paper are not affected. We apologize for the inconvenience.

In our manuscript, we first selected well-operating elements from a library containing ID and NOT elements and all possible permutation and orientation of integrase attachment sites. To speed up the process of characterization, we directly synthesized all constructs corresponding to the element before and after recombination occurred. Results for these characterizations are presented in the revised paper in Figure 2, Figure S2 and S3, and S4.

According to their orientation of attB and attP sites, the excision reaction can lead to a remaining attL or attR site. In the original presentation of the data, we inverted the resulting site for BR-PR and PR-BR recombinations (Fig. A below).

Figure A. Misattribution of recombination products and corrected version.

This change results in a modification of the characterization plot (Now presented in Fig S2 and S3 of the manuscript) as presented in Figure B below.

Figure B. Example of changes in element characterization plot after correction.

Basically, the values for the recombination products resulting from BR-PR and PR-BR are inverted. This results in a modification of the calculated fold changes between ON and OFF states for those particular sites configurations only. The data concerning BF-PF and PF-BF are not affected at all.

A table summarizing the different values of the fold changes before and after correction is presented below.

Table 1: comparison between the calculated fold changes between ON and OFF states before (red) and after (blue) correction. Fold changes were calculated using data from the supplementary file (“Source data”).

ID-element	Bxb1	Tp901	Int5	Int7
ID-BF-PF	35,1	5,7	12,4	132
ID-BR-PR	184 / 308	3,4 / 1,7	229 / 58,0	257 / 114
ID-PF-BF	7,2	3,2	316	270
ID-PR-BR	371 / 222	3,8 / 7,5	2,4 / 9,6	288 / 649

First version / Corrected

Selected element

NOT-element	Bxb1	Tp901	Int5	Int7
NOT-BF-PF	102	35,7	517	626
NOT-BR-PR	413 / 378	35,4 / 269	766 / 761	205 / 482
NOT-PF-BF	131	261	418	178
NOT-PR-BR	565 / 620	292 / 38,5	405 / 407	559 / 238

Some selected elements are not affected by the correction and remain the best choice (e.g. Int5-ID-PF-BF). Even with a corrected fold change, some of the selected elements are still the best of all 4 possibilities (e.g. Int7-ID-PR-BR, Bxb1-NOT-PR-BR) or with a value

comparable to the element presenting the highest fold change (e.g. Bxb1-ID-PR-BR). No significant modification is observed for the TP901-ID-elements for which we needed to change the terminator. The most significant change is observed for Int7-NOT-elements. However, the selected element is still behaving correctly and is present in only one device in Figure 3 which works well in response to this input (input D for gate [NOT(A). NOT(B) . NOT(C). NOT (D)]).

In all, the selected elements still qualify for “well-operating”, with a fold change comprised between 83 and 649. We have redone the figures, amended the text and supplementary materials to reflect the changes associated with this slight change in the data.

The main message of our paper remains unaffected:

- For a given logic element, different orientations and positions of recombination sites can affect downstream gene expression.
- Systematic screening of all combinations is required in order to identify well operating elements (high-fold change, low fluorescence in their OFF state).
- Those well-operating elements can be combined to predictably assemble logic devices working reliably (Fig4 of the revised main text).

We apologize again for this error and the inconvenience it causes in the review process.

Reviewers' Comments:

Reviewer #1:

Remarks to the Author:

The authors have, in my opinion, addressed all points raised in mine and reviewer #2's comments.

The correction described in the response does not affect interpretation of the data and a sufficient explanation has been provided.

Minor comments:

Grammatical correction are needed in line 59, 134 and 516.

REVIEWERS' COMMENTS:

Reviewer #1 (Remarks to the Author):

The authors have, in my opinion, addressed all points raised in mine and reviewer #2's comments.

The correction described in the response does not affect interpretation of the data and a sufficient explanation has been provided.

Minor comments:

Grammatical correction are needed in line 59, 134 and 516.

Thank you for the comment, we have corrected the errors.